# COVID-19-Related Coagulopathy—Is Transferrin a Missing Link?

**DOI:** 10.3390/diagnostics10080539

**Published:** 2020-07-30

**Authors:** Katie-May McLaughlin, Marco Bechtel, Denisa Bojkova, Christian Münch, Sandra Ciesek, Mark N. Wass, Martin Michaelis, Jindrich Cinatl

**Affiliations:** 1School of Biosciences, University of Kent, Canterbury CT2 7NJ, UK; km625@kent.ac.uk; 2Institute for Medical Virology, University Hospital, Goethe University, 60596 Frankfurt am Main, Germany; marco.bechtel94@gmx.de (M.B.); Denisa.Bojkova@kgu.de (D.B.); Sandra.ciesek@kgu.de (S.C.); 3Institute of Biochemistry II, Faculty of Medicine, Goethe University, 60590 Frankfurt am Main, Germany; ch.muench@em.uni-frankfurt.de; 4Frankfurt Cancer Institute, Goethe University, 60590 Frankfurt am Main, Germany; 5Cardio-pulmonary Institute, Goethe University, 60590 Frankfurt am Main, Germany; 6German Center for Infection Research, DZIF, External Partner Site, 60590 Frankfurt am Main, Germany; 7Fraunhofer Institute for Molecular Biology and Applied Ecology (IME), Branch Translational Medicine und Pharmacology, 60590 Frankfurt am Main, Germany

**Keywords:** SARS-CoV-2, COVID-19, thrombosis, coagulation, coagulopathy, transferrin

## Abstract

SARS-CoV-2 is the causative agent of COVID-19. Severe COVID-19 disease has been associated with disseminated intravascular coagulation and thrombosis, but the mechanisms underlying COVID-19-related coagulopathy remain unknown. The risk of severe COVID-19 disease is higher in males than in females and increases with age. To identify gene products that may contribute to COVID-19-related coagulopathy, we analyzed the expression of genes associated with the Gene Ontology (GO) term “blood coagulation” in the Genotype-Tissue Expression (GTEx) database and identified four procoagulants, whose expression is higher in males and increases with age (ADAMTS13, F11, HGFAC, KLKB1), and two anticoagulants, whose expression is higher in females and decreases with age (C1QTNF1, SERPINA5). However, the expression of none of these genes was regulated in a proteomics dataset of SARS-CoV-2-infected cells and none of the proteins have been identified as a binding partner of SARS-CoV-2 proteins. Hence, they may rather generally predispose individuals to thrombosis without directly contributing to COVID-19-related coagulopathy. In contrast, the expression of the procoagulant transferrin (not associated to the GO term “blood coagulation”) was higher in males, increased with age, and was upregulated upon SARS-CoV-2 infection. Hence, transferrin warrants further examination in ongoing clinic-pathological investigations.

## 1. Introduction

Severe acute respiratory syndrome coronavirus 2 (SARS-CoV-2) is the causative agent of the ongoing coronavirus disease 2019 (COVID-19) outbreak [1,2]. SARS-CoV-2 was first detected in December 2019 in the Chinese city Wuhan and has since spread around the world. The John Hopkins University Coronavirus Resource Center [3] currently reports more than 16 million confirmed COVID-19 cases and more than 650,000 confirmed COVID-19-related deaths. The majority of SARS-CoV-2 infections are mild with a substantial proportion of asymptomatic cases. However, SARS-CoV-2 causes in some patients severe life-threatening multi-organ disease [1,4,5,6].

Severe COVID-19 disease has been associated with disseminated intravascular coagulation and thrombosis [1,4,5,6], but the mechanisms underlying COVID-19-related coagulopathy remain unknown. It is known, however, that the risk of severe and fatal COVID-19 disease is higher in males than in females and that it increases with age [7]. Similarly, the risk of coagulation-related pathologies and thrombosis increases with age and is further enhanced in males [8,9]. Thus, gene products that (1) are involved in coagulation, (2) change with age, (3) differ in their levels between females and males, and (4) are regulated in response to SARS-CoV-2 infection represent candidate factors that may contribute to COVID-19-related coagulopathy and disease severity.

To identify such candidate factors that may be involved in COVID-19-related coagulopathy, we here performed a combined analysis of a proteomics dataset derived from SARS-CoV-2-infected cells [10], of a dataset of host cell proteins found to bind to SARS-CoV-2 proteins [11], and of human gene expression data from the Genotype-Tissue Expression (GTEx) database [12].

## 2. Materials and Methods

### 2.1. Data Acquisition

Gene Ontology is an initiative that annotates genes with functions [13]. Genes associated with the Gene Ontology (GO) term “Blood Coagulation” (GO:0007596) were identified using the online database AmiGO 2 [13]. This generated a list of 335 unique genes annotated with 23 unique terms (including “Blood Coagulation” and 22 child terms) for further analysis.

Gene expression data (transcripts per million, TPM) and clinical data for 980 individuals (17,382 samples from 30 tissues) were downloaded from the GTEx Portal (https://www.gtexportal.org/home/datasets; GTEx Project, version 8). We also used normalized protein abundance data from a recent publication [10] in which protein abundance in uninfected and SARS-CoV-2-infected Caco-2 (SARS-CoV-2-susceptible colorectal cancer cell line) cells was quantified. Data were subsequently normalized using summed intensity normalization for sample loading, followed by internal reference scaling and Trimmed mean of M normalization.

We also queried the EBI IntAct database (https://www.ebi.ac.uk/intact/) for “annot:dataset“coronavirus””, and filtered for interactions between human proteins and SARS-CoV-2 proteins. These SARS-CoV-2-interacting proteins were derived from a study by Gordon et al. [11], in which 29 SARS-CoV-2 proteins were cloned, tagged, and expressed in HEK293T cells, and incubated for 40 h prior to affinity purification and identification of binding partners by mass spectrometry. This resulted in 332 high confidence human protein interactors for 26 of the SARS-CoV-2 proteins.

### 2.2. Data Analysis

Analyses were performed using R3.6.1. Linear models were generated to estimate the relationship between gene expression and age using the base R function *lm*, which generated *p*-values indicating the significance of the relationship. Models with a *p*-value <0.05 were considered significant. Plots were generated using the R package *ggplot2*. Mean protein abundance (for proteomics data) and median gene expression TPM (for GTEx data) was plotted using the function *ggviolin*. *p*-values indicating the significance of the difference between gene expression/protein abundance in males and females in each given age group were the result of a Wilcoxon rank sum test for independent groups. For the proteome data, we performed a two-sided student’s *t*-test. Boxplots comparing gene expression in males and females were generated using the function *ggboxplot*, for which *p*-values were the result of a Wilcoxon rank sum test for independent groups.

## 3. Results

### 3.1. Identification of Genes that May Be Associated with an Increased Coagulation Risk in Males and at an Older Age

Using AmiGO 2 [12], we identified 335 genes, which are associated with the GO term “blood coagulation” (GO:0007596). Since the risk of severe COVID-19 disease increases with age and is higher in males than in females [7], coagulation-associated genes which may be relevant in the context of COVID-19-related coagulopathy would be expected to differ in their expression between females and males and change in their expression with age. 

An analysis of the genes associated with the GO term “blood coagulation” using the Genotype-Tissue Expression (GTEx) database [12] resulted in 256 coagulation-associated genes, that are differently expressed between females and males (Appendix A) and 237 genes whose expression changed with age (Appendix A). These lists included many genes, whose products are involved in the regulation of upstream processes, which may be linked to coagulation in certain cell types and under certain circumstances but are not core players directly involved in the actual coagulation process. Examples include members of major signaling cascades such as PI3K or MAPK signaling (Appendix A). Hence, the functions of these genes were manually annotated to identify candidate genes, whose products act as procoagulants and anticoagulants (Appendix A). This resulted in a list of 49 overlapping genes, whose products are directly involved in coagulation (Appendix A).

Two groups of genes were considered as candidates, whose products may increase or reduce the risk of COVID-19-related coagulopathy: (1) procoagulants that display higher expression in males and increase in their expression with age and (2) anticoagulants that display higher expression in females and decrease in their expression with age. According to these criteria, we found four procoagulants (ADAMTS13, F11, HGFAC, KLKB1) and two anticoagulants (C1QTNF1, SERPINA5), whose expression may predispose males and older individuals to COVID-19-related coagulopathy and severe COVID-19 disease (Table 1, Figure 1, Appendix A).

### 3.2. No Overlap between COVID-19-Related Coagulopathy Predisposition Genes and SARS-CoV-2-Associated Genes

Next, we investigated whether there is a known relationship between the candidate genes, whose products may predispose individuals to severe COVID-19 disease, and SARS-CoV-2 infection. For this, we used a proteomics and translatome dataset derived from SARS-CoV-2-infected and non-infected cells [10] (Appendix A) and 332 high confidence human SARS-CoV-2 interactor proteins, which had been identified by expressing 29 tagged SARS-CoV-2 proteins in HEK293T cells, followed by affinity purification and mass spectrometric identification of binding partners [11] (Appendix A). However, none of our six candidates were shown to be regulated by or interact with SARS-CoV-2 (Appendix A).

### 3.3. Transferrin May Be Involved in COVID-19-Related Coagulopathy

While searching manually for additional candidates potentially involved in COVID-19-related coagulopathy, we found transferrin to be upregulated in SARS-CoV-2-infected cells relative to non-infected cells [10] (Figure 2A). Transferrin is a glycoprotein circulating in the blood that is best known for its function as an iron carrier. It binds to cellular transferrin receptors and delivers iron by receptor-mediated endocytosis [14,15]. However, transferrin has also been shown to increase coagulation independent of its role as an iron transporter by interfering with antithrombin/SERPINC1-mediated inhibition of coagulation proteases including thrombin and factor XIIa [16]. Hence, there might be a link between transferrin levels and coagulation in COVID-19 patients.

GTEx data indicated that transferrin expression increased with age and was higher in males than in females (Figure 2B). In contrast, expression of its antagonist antithrombin did not increase with age and was similar in females and males (Figure 2C). Thus, the transferrin/antithrombin ratio increases with age and is higher in males than in females (Figure 2D). This correlates with the risk of severe and fatal COVID-19 disease, which is higher in males than in females and also increases with age [7]. Hence, an increased transferrin/antithrombin ratio may contribute to COVID-19-related coagulopathy and more severe disease in older patients, in particular in males.

Transferrin was not included in the list of SARS-CoV-2-interacting proteins [11] (Appendix A). This suggests that transferrin is regulated in response to SARS-CoV-2 infection but does not directly interact with SARS-CoV-2 proteins.

## 4. Discussion

Severe COVID-19 disease is associated with intravascular coagulation and thrombosis (COVID-19-related coagulopathy) [1,4,5,6,17] and the risk of severe disease increases with age and is higher in males than in females [7]. To identify factors involved in coagulation that may contribute to COVID-19-related coagulopathy, we used genes associated with the GO term “blood coagulation” and the GTEx database resulting in four procoagulants, whose expression was higher in males than in females and increased with age (ADAMTS13, F11, HGFAC, KLKB1), and two anticoagulants, whose expression was higher in females and decreased with age (C1QTNF1, SERPINA5).

However, these candidate factors were not found to be regulated in SARS-CoV-2-infected cells [10] or among proteins known to interact with SARS-CoV-2 proteins [11]. Hence, they may rather generally predispose individuals to coagulopathy, which is known to be higher in males and to increase with age [8,9], than being directly involved in the disease processes mediated by SARS-CoV-2 infection. Therefore, these factors may also be relevant in age- and gender-related thrombosis formation beyond COVID-19.

Our manual further investigations identified transferrin as a candidate factor, which may be involved in COVID-19-related coagulopathy. The expression of transferrin, a known procoagulant [16], was upregulated in SARS-CoV-2-infected cells, increased with age, and was higher in males than in females. Moreover, a rise of transferrin levels was observed in patients during COVID-19 disease progression [18]. Transferrin is an iron carrier protein that circulates and delivers iron to cells via transferrin receptor binding followed by receptor-mediated endocytosis [14,15]. However, it also promotes coagulation by iron-independent mechanisms as an inhibitor of antithrombin, which interferes with the prothrombotic activity of coagulation proteases such as thrombin and factor XIIa [16].

Transferrin is primarily produced in the liver. However, (SARS-CoV-2-induced) locally produced transferrin may contribute to COVID-19 pathology, even independent of circulating transferrin levels [14,15,19,20,21,22,23,24]. For example, transferrin is produced in the brain [22], and high transferrin levels have been associated with hypercoagulability and ischemic stroke [22]. Stroke is a significant complication in COVID-19 [25] and is much more common in COVID-19 than, for example, in influenza patients [26]. Both ischemic and hemorrhagic strokes are observed in COVID-19 patients [25]. Notably, transferrin may not only contribute to ischemic strokes via inducing coagulation [27], it may also increase the brain injury associated with hemorrhagic strokes by facilitating cellular iron uptake [28]. High transferrin levels have also been associated with diabetes and metabolic syndrome [23,29,30,31], which are known risk factors for severe COVID-19 disease [32,33,34].

## 5. Conclusions

In conclusion, the role of transferrin in the course of COVID-19 disease and in particular of COVID-19-related coagulopathy should be considered and further examined in ongoing clinico-pathological investigations. If the role of transferrin is confirmed in the pathogenesis of severe COVID-19 disease and in COVID-19-related coagulopathy, it is a candidate diagnostic marker for the monitoring of COVID-19 progression and may guide the use of anticoagulants in COVID-19 patients.

## Figures and Tables

**Figure 1 diagnostics-10-00539-f001:**
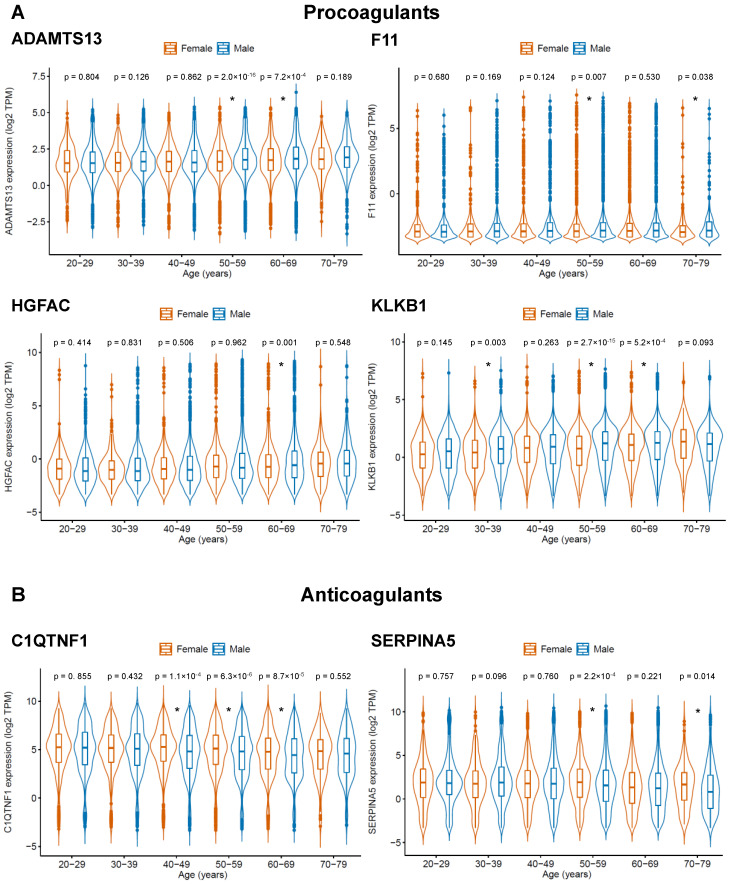
Gene products anticipated to be of potential relevance for COVID-19-related coagulopathy, based on genes with a role in coagulation that are differentially expressed between females and males (Appendix A) and whose expression correlates with age (Appendix A). Candidate gene products were either (**A**) procoagulants (ADAMTS13, F11, HGFAC, KLKB1), which display higher expression in males than in females and increase with age, or (**B**) anticoagulants (C1QTNF1, SERPINA5), which display higher expression in females than in males and decrease with age. A complete list of the relevant genes overlapping between Appendix A and Appendix A is presented in Appendix A. *p*-values were determined by two-sided Student’s *t*-test. * *p*-value < 0.05.

**Figure 2 diagnostics-10-00539-f002:**
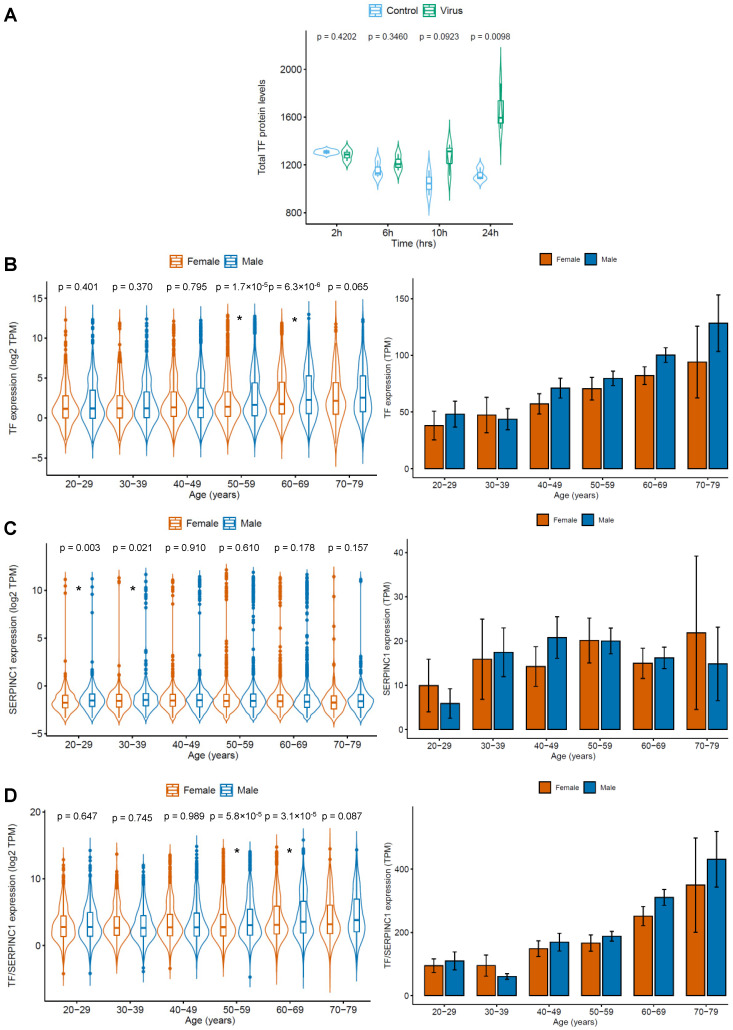
SARS-CoV-2-induced (derived from [9]) transferrin (TF) expression and age- and gender-specific expression of the procoagulant TF and its antagonist SERPINC1/antithrombin based on GTEx data. Data are presented as violin blots to indicate the distribution of individual values and as bar charts to facilitate comparisons. TF displayed higher levels in SARS-CoV-2-infected cells than in non-infected cells. Moreover, TF expression and the expression ration of TF and SERPINC1 increased with age and were higher in females than in males. (**A**) TF protein abundance in uninfected (control) and SARS-CoV-2-infected (virus) Caco-2 cells. *p*-values are the result of a two-sided Student’s *t*-test. (**B**) TF expression (TPM) in females and males across six age groups. *p*-values were calculated using the Wilcoxon rank sum test for independent groups. (**C**) SERPINC1 expression (TPM) in females and males across six age groups. *p*-values were calculated using the Wilcoxon rank sum test for independent groups. (**D**) Ratio of TF/SERPINC1 expression (TPM) in females and males across six age groups. *p*-values were calculated using the Wilcoxon rank sum test for independent groups. * *p*-value < 0.05.

**Table 1 diagnostics-10-00539-t001:** Candidate gene products that may be involved in COVID-19-related coagulopathy. Candidates were either (**A**) procoagulants (ADAMTS13, F11, HGFAC, KLKB1), which display a lower expression in females than in males and increase with age, or (**B**) anticoagulants (C1QTNF1, SERPINA5), which display higher expression in females than in males and decrease with age.

**(A) Procoagulants**				
	Female vs. male		Age-associated expression	
	Relative expression	*p*-value *	Direction	*p*-value
ADAMTS13	low	8.9 × 10^−8^	increase	1.6 × 10^−11^
F11	low	3.5 × 10^−4^	increase	<2.2 × 10^−16^
HGFAC	low	0.032	increase	0.049
KLKB1	low	<2.2 × 10^−16^	increase	4.3 × 10^−6^
**(B) Anticoagulants**				
	Female vs. male		Age-associated expression	
	Relative expression	*p*-value	Direction	*p*-value
C1QTNF1	high	6.7 × 10^−13^	decrease	2.6 × 10^−16^
SERPINA5	high	3.7 × 10^−3^	decrease	2.8 × 10^−6^

* *p*-value < 0.05 were considered as significantly different.

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
