# Peer review of "COVID-19-Related Coagulopathy—Is Transferrin a Missing Link?"

_diagnostics, 2020, doi:10.3390/diagnostics10080539_

Round 1
Reviewer 1 Report
The manuscript “COVID-19-related coagulopathy – Is transferrin a missing link?” by McLaughlin et al. explores COVID-19 coagulopathy biomarkers. It is nice to see manuscripts that test new hypotheses using publicly available data sets. It is important to continue to build on pervious work and explore new links, especially with high-throughput data sets where there is much to be found. This manuscript is repetitive, but well written. My major concern is that the conclusions are weak, the majority of the comparisons do not have significant p-values and the plots do not demonstrate strong relationships. It would also be beneficial if some experimental work was completed to add more support to the conclusions.
Introduction:
An introduction to transferrin could be added. This manuscript is a bit repetitive overall; I feel that much of what is in the introduction is then restated in the discussion. This should be better balance between the two sections.
Results:
Figure 1: Asterisks should be used to mark what comparisons are significant. These plots would also be easier to read if the pro and anticoagulants were grouped, perhaps separated into A) and B). In general, these trends are not very apparent, only a few of the p-values are significant. It is not very convincing.
Discussion:
There isn’t much interpretation to the results, such as the mechanism behind the exact role transferrin plays.
Reviewer 2 Report
The manuscript is absolutely interesting since it contributes to the understanding of factors influencing COVID-19 infections. However, I have some concerns that should be addressed:
-Introduction should be enlarged since it not completely provide the state of the art with respect to the topic.
-other point, the graphs should be edited in order to help the readers with the discussion since in this form is really hard to understand the differences of expression in male/female and for different ages (with exclusion of Fig 2A). I suggest to introduce novel graphs maybe based on histograms or related representations for really observing the changes in the expression.
-Conclusion, the section should be enlarged focusing the discussion on the possible role of TF as diagnostics, this is necessary due to the aim of the journal.
After this changes the manuscript can be re-evaluated.
Round 2
Reviewer 1 Report
The manuscript is much improved.
My only minor comment is that Figure 2 should also have significant p-values marked with an asterisk for each comparison.
Author Response
This was done.
Reviewer 2 Report
The authors addressed all my previous concerns. Accordingly, I recommend the publication of the article.
Author Response
Many thanks.